# A multi-scale clutch model for adhesion complex mechanics

**Chiara Venturini[1], Pablo Sáez[1,2,3]** *

**1** Laboratori de Càlcul Numèric (LaCaN), Universitat Politècnica de Catalunya, Barcelona, Spain, **2** E.T.S. de Ingeniería de Caminos, Universitat Politècnica de Catalunya, Barcelona, Spain, **3** Institut de Matemàtiques de la UPC-BarcelonaTech (IMTech), Universitat Politècnica de Catalunya, Barcelona, Spain

* pablo.saez@upc.edu

**Data Availability Statement:** The source code used to produce the results presented in this manuscript is available on Zenodo at link: https://doi.org/10.5281/zenodo.7906839. Extended figures are uploaded in https://doi.org/10.5281/zenodo.7907673.

## Abstract

Cell-matrix adhesion is a central mechanical function to a large number of phenomena in physiology and disease, including morphogenesis, wound healing, and tumor cell invasion. Today, how single cells respond to different extracellular cues has been comprehensively studied. However, how the mechanical behavior of the main individual molecules that form an adhesion complex cooperatively responds to force within the adhesion complex is still poorly understood. This is a key aspect of cell adhesion because how these cell adhesion molecules respond to force determines not only cell adhesion behavior but, ultimately, cell function. To answer this question, we develop a multi-scale computational model for adhesion complexes mechanics. We extend the classical clutch hypothesis to model individual adhesion chains made of a contractile actin network, a talin rod, and an integrin molecule that binds at individual adhesion sites on the extracellular matrix. We explore several scenarios of integrins dynamics and analyze the effects of diverse extracellular matrices on the behavior of the adhesion molecules and on the whole adhesion complex. Our results describe how every single component of the adhesion chain mechanically responds to the contractile actomyosin force and show how they control the traction forces exerted by the cell on the extracellular space. Importantly, our computational results agree with previous experimental data at the molecular and cellular levels. Our multi-scale clutch model presents a step forward not only to further understand adhesion complexes mechanics but also to impact, e.g., the engineering of biomimetic materials, tissue repairment, or strategies to arrest tumor progression.

## Author summary

Cell-matrix adhesions are directly implicated in key biological processes such as tissue development, regeneration, and tumor cell invasion. This cell function is determined by how adhesion complexes feel and respond to mechanical forces. Still, how forces are transmitted through the individual cell adhesion molecules that integrate the adhesion complex is poorly understood. To address this issue, we develop a multi-scale clutch model for adhesion complexes where individual adhesion chains, made of integrin and talin

**Funding:** We acknowledge funding from the Spanish Ministry of Science and Innovation (Grant PID2019-11094GB-100 funded by MCIN/ AEI /10.13039/501100011033 to P.S.), the European Commission (H2020-FETPROACT-01-2016-731957 to C.V. and P.S) the Generalitat de Catalunya (2017-SGR-1278 to P.S. and FI AGAUR 2018 for C.V. salary). The funders had no role in study design, data collection and analysis, decision to publish, or preparation of the manuscript.

**Competing interests:** The authors have declared that no competing interests exist.

molecules, are considered within classical clutch models. This approach provides a rich mechanosensing insight into how the mechanics of cell adhesion works. It allows the integration of accurate biophysical models of individual adhesion molecules into whole adhesion complex models. Our multi-scale clutch approach extends the current knowledge of adhesion complexes and also impacts current strategies for tissue regeneration, control of tumor progression, and engineering biomimetic materials.

## Introduction

Cell adhesions are central to maintaining the right function of the cell [1]. Specifically, integrin-based cell adhesions are responsible for the attachment of cells to the extracellular matrix [2, 3] and are crucial for single and collective cell motility in development and disease [4]. We focus here on integrin-based cell adhesions, and we refer to this type of adhesion complexes (ACs) when we next talk about cell adhesion. ACs are made of an assembling of Cell Adhesion Molecules (CAMs) [5] linked together to establish a rich mechanosensitive and mechanotransductive system that regulates cell adhesion and function [6, 7]. Moreover, ACs grow and respond differently to a number of factors, including the rigidity of the Extracellular Matrix (ECM) [8, 9] and the contractile forces exerted by actomyosin networks [10]. Today, our knowledge of how ACs respond to these factors is extensive thanks to mounting experimental evidence. In the remaining introductory section, we review the organization and the mechanical behavior of the ACs and their regulation by the ECM.

### Force distribution and architecture in integrin-based cell adhesions

At the nanoscale, ACs are made of ECM ligands, transmembrane and cytoplasmatic proteins, and a network of actin filaments and myosin motors (see Fig 1). The hierarchical composition of the AC has been analyzed by measuring the kinematics of each molecule by fluorescence speckle microscopy [11]. Integrins are weakly correlated with the actin flow, indicating that they should be attached to the immobilized ECM. On the other hand, $\alpha$-actinin correlated with the actomyosin network, suggesting that it is fully interconnected within the actomyosin network. Vinculin and talin showed a partial coupling to the actin flow, which indicated that these components connect the actomyosin structures to the immobile extracellular binding sites. The relation of these components with the actomyosin flow suggested a stratified organization of the main adhesion molecules (Fig 1). Three-dimensional super-resolution fluorescence microscopy confirmed this organization of CAMs and provided a more precise description of the nanoscale architecture of the AC [5]. They showed that integrins and actin are separated vertically $\approx$ 40 nm by a first signaling layer consisting of the integrin cytoplasmatic tails, a second layer of talin and vinculin, oriented at 15° with respect to the plasma membrane [12], and an actin regulatory layer containing zyxin and $\alpha$-actinin, among others, connecting talin with the actin network. Therefore, these adhesion molecules lie mostly parallel to the cell-ECM contact plane. The link between integrins, talin, vinculin, and the actomyosin network is the most determining link of CAMs in cell-ECM adhesion [7].

In the inner cellular layer, the *actomyosin network* is made of a template of 10 to 30 actomyosin filaments, linked periodically through $\alpha$-actinin and non-muscle myosin motors that fuse to finally form contractile stress fibers [7, 13–15]. Myosin motors generate a pulling force that has been implicated in the maturation [7, 16, 16], stability [10, 17] and disassembling [7, 16] of ACs. The force exerted by the myosin motors causes conformational changes in the main CAMs, such as talin and vinculin. Although it is now clear that myosin II plays a key role in

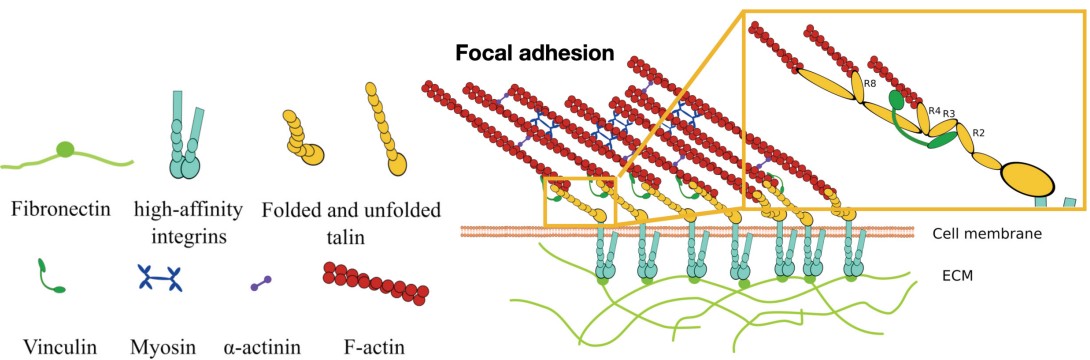

**Fig 1. Composition and organization of an AC.** (a) the main cell components of the AC: fibronectin, transmembrane integrins, talin, vinculin, and an actomyosin network made of F-actin, myosin motors, and $\alpha$-actinin. (b) Organization of these adhesion complex components in the AC. Fibronectin, laminin, or collagen serve as anchoring points for the transmembrane integrins to attach to the extracellular space. Integrins bind to the talin rod, which is also connected to the actin network through specific actin binding sites. Actin filaments bundle together mediated by $\alpha$-actinin to form stress fibers and exert forces on the adhesion chain by the active pulling forces of myosin motors.

the active force generation in cell adhesions [10, 18], how it controls the adhesion dynamics is not completely understood.

The second signaling layer is made of talin and vinculin. *Talin* is recruited from the cytosol to the adhesion site [19] and binds to the intracellular domain of integrins, the $\beta$-integrin unit, from one side, and to the actin cytoskeleton [20, 21] from its most-inner domain. The force exerted by the actomyosin fibers induces talin deformation and conformational changes [22–24] (more details on the talin structure and mechanics are discussed below). Upon unfolding of the talin rod domains, *vinculin* binds to the exposed talin domains through Vinculin Binding Sites (VBSs) [25]. The binding rates of vinculin to talin increase upon stretching of the talin rod [23, 24].

In the outer molecular layer, transmembrane *integrin* molecules connect the cell to the ECM [3, 26], via laminin, fibronectin, or collagen. Integrins diffuse freely within the nascent adhesion. Because the cytoplasmic domains of integrins cannot directly bind to actin, integrins use talin and kidlin as intermediate elements to connect to the actin filaments. Low-affinity integrins are activated by their attachment to talin, when integrins switch from a low-affinity state, where they appear bent and closed, to a high-affinity state [27–31]. Then, upon force-induced activation [19], integrins change again their structure passing to an extended and open state. Integrin activation fosters the binding of integrins with the ECM [32], mediated by talin [33, 34] and vinculin signaling [35]. Eventually, integrins cluster in the adhesion complex and the diffusivity of the integrins outside the adhesion reduces, which, eventually, leads to the formation of stable ACs [36].

Force transmission through all these CAMs is central to the cell adhesion behavior. Force estimations at single integrins have been reported in $\approx$2–100 pN [3, 29]. Most of these measures were obtained by averaging across the entire adhesion plaque, which only provides an estimation of the actual forces experienced by single integrins. For example, a force value of 1–2 pN at individual integrins was calculated in adhered fibroblasts by estimating the number of bound integrins per unit area [37]. Similarly, a force of $\approx$10 pN was obtained from traction forces of $\approx$1 kPa in $\approx$ 500 integrins / $\mu$m$^2$ [9]. FRET-based molecular sensors pinpointed the single integrin force in 1–5 pN [38, 39]. DNA-based sensors established a force range of 33–43 pN [28] in $\alpha_V\beta_3$ integrins, one of the most ubiquitous integrins across cell types. A detailed

analysis of the lifetime of the $\alpha_V\beta_3$-fibronectin bond showed a maximum lifetime of the bond at 30 pN, vanishing above $\approx$60 pN [40], confirming values from DNA-based sensors. Measures of the force sensed by talin [41] were pinpointed to 7–10 pN and the limit force when all domains are completely unfolded is 30 pN [24]. All these data suggest a narrow mechano-sensing force of few pN across the adhesion chains.

Moreover, because the talin-integrin stoichiometry is approximately 1:1 in ACs [5, 42], an integrin-talin link connected in series and stretched from one end shares the same force. In this situation, a 30 pN force would unfold all domains of the talin rod [24], an extreme case of the talin state, while the lifetime of the integrin-fibronectin catch bonds would be maximum and, therefore, the force could still increase across the molecular chain.

In summary, there are differences in the force magnitudes measured on these molecules. Whether these differences are real and biologically designed for the right function of the cell, or they are just measuring errors, is not yet clear.

## The extracellular matrix architecture as a regulator of cell adhesion

The stiffness of the ECM and the spacing of the specific binding sites represent the most determining factors of the ECM for AC behavior. The ECM rigidity directly determines the force transmission from the active contraction generated by the actomyosin network through the CAMs. The cell adhesion behavior as a function of the substrate rigidity was described by the motor-clutch hypothesis [43–46]. When integrins bind to rigid substrates, myosin contractility builds force very quickly in the molecular chain and the CAMs must accommodate the imposed displacements. Unbinding rates become faster than binding rates, which limits force transmission to stiff substrates, and the AC completely disengages before other ligands have time to bind. This behavior has been termed "frictional slippage" [43, 46]. On the other hand, soft substrates deform substantially upon force application, which induces a low force transmission to the CAMs and a cooperative engagement of many bonds over time. As ligands keep binding, myosin contractility deforms the substrate, increasing the force transmission to the substrate, until the bonds reach the limit rupture force. At that point, the AC disassembles, and the cycle starts over. This behavior was previously termed "load-and-fail" dynamics [43, 46].

The AC behavior also depends on the adhesion-site patterning [9, 47]. The ECM is made of proteins, such as fibronectins and laminins, that serve as binding sites for cells to attach. In low ligand spacing and stiff substrates, the maturation of ACs is highly impaired, while in soft substrates ACs consistently mature into long-lived focal adhesions. However, when ligands are put apart by more than a few hundred nanometers, focal complexes adhesion formation is impaired in both soft and stiff substrates [47, 48], probably due to a restriction in integrin clustering, suggesting an adaptor ruler of the order of nanometers that could control cell adhesion maturation. It has also been suggested that the local number and spacing, and not the global density or distribution of ligands, are sufficient to induce adhesion mechanisms [49].

Further research on the cooperative response of ligand spacing and the ECM rigidities in cell adhesion showed contra-intuitive results [50]. On soft substrates, only small nascent adhesions formed for ligand spacing of 50 nm and 100 nm. For these spacing, AC formation was observed only for substrate rigidities above 5 kPa and below 100 kPa, although it was not seen in the glass. For 200 nm spacing, ACs were observed to form below substrates of 5 kPa. Although most of these results could be described through a ruler mechanism, the formation of ACs for a spacing larger than 200 nm on substrates of $\approx$ 1 kPa, could not be explained by that hypothesis.

In short, it is clear that the spacing of ligands and the rigidity of the ECM determine cell behavior. However, how specifically the cooperative mechanical response of each CAM to these ECM aspects maps to the whole adhesion complex behavior is still not well understood.

## Models for cell adhesion and goal of this work

Different physical models have been proposed in the literature to rationalize cell adhesion mechanics. Continuum models have been used to describe the clustering and growth of ACs [51–55], and the role of the ECM [56] and of cell contractibility, [53, 54] in adhesion mechanics. Discrete models have been also used to model cell-ECM mechanics [57–59]. Coarse-grained models based on Brownian Dynamics also show that the morphology and distribution of nascent adhesions depend on the nanoscale ligand affinity of $\beta$-1 and $\beta$-2 integrins [60]. Other approaches use repeated random sampling simulations, mostly through Monte Carlo (MC) and Gillespie methods, to model the dynamics of cell adhesion [61–63].

In this framework, the Molecular Clutch model [8, 9, 43] describes the specific binding/ unbinding events of individual molecular chains, or clutches, and provides a simple but insightful explanation of cell adhesion mechanics as a function of the substrate rigidity. The clutch models relate the velocity of the retrograde flow of the cell, generated by the pulling forces of myosin motors, with the mechanical properties of the ECM and the clutches dynamics. It was first developed to reproduce the "load and fail" of adhesion in the growth cone of a neuron [9, 43]. The model has been further exploited since then, providing remarkable insights and qualitative understanding of how cell adhesions form, behave, and modify the whole cell mechanics [8, 64–68]. A detailed description of the model is presented in the Methods section. The molecular clutch model successfully reproduces the adhesion behavior at the cell scale, specifically the cell traction $P$ and the actin velocity $v$, for different cell types (see e.g. [8, 43, 46, 69] and section *Results of the clutch model* in the S1 Text). Therefore, it represents an excellent option when we study the adhesion behavior of an entire cell. However, when we look at the nanoscale, we see a number of simplifications introduced in the modeling of the adhesion structure. The clutch models consider a single linear spring for the substrate, which naturally results in one single quantity for the substrate displacement. Considering a substrate with nano-patterned attaching locations where integrins adhere, the displacement of the substrate should change point-wise, depending on the locations of the ligands. This issue has been tackled by including a number of spring elements to represent the ECM [50]. Previous clutch models have simplified the behavior of the chain of CAMs to a linear spring. Integrin and talin, which are key components of the AC [11, 70], are not explicitly incorporated but have a distinctive mechanical response [3, 24] that contributes significantly to the behavior of the AC. For example, the talin rod undergoes folding and refolding dynamics of all its domains, dramatically changing the force and displacement of the adhesion chain. Moreover, each domain shows a non-linear force-displacement response to load [24]. These model simplifications may explain some contradicting results between model variables at the molecular scale and previous experimental data. Specifically:

- We analyzed the deformation of each clutch and compare it to the actual behavior of individual CAMs. Integrins in their bent and closed configuration, or low-affinity state, have a length of $\approx$ 11–13 nm [31, 71]. Upon activation, they pass to an extended and open configuration, a high-affinity state, with a length of 18–23 nm [12, 71, 72], from which they can bind and create focal adhesions. $\alpha_5\beta_1$ integrins can reach a total length of 50 nm, therefore integrins displacement can reach $\approx$ 30 nm [31, 71, 73]. Similarly, the talin rod is $\approx$ 60–80 nm in length [12, 74, 75] in its open, fully folded state and its end-to-end length when fully unfolded under force reaches $\approx$ 800 nm [24]. This gives a total displacement up to $\approx$ 730 nm. In the model results, the displacement of the molecular chain has a maximum value of $\approx$ 2 nm (Fig A1-A2 in S1 Text). This displacement is two orders of magnitude smaller than the possible displacements of the integrin and talin molecules.

- Forces have been measured in individual integrins ($0 - 40$ pN) and talins ($0 - 30$ pN). We plot the mean value and standard deviation of the force as a function of the stiffness of the substrate and the number of ligands $n_c$ (see Fig A7 in S1 Text). We see that the forces remain similar for all three values of the substrate rigidity, up to $\approx 60$ pN (Fig A1-A2 and Fig A7 in S1 Text). Although these values are closer to those reported at the individual molecular level, a force of 60 pN would completely unfold all talin domains, which represents an extreme case of the talin behavior.

In summary, current clutch models simplify the actual molecular organization of ACs, which hinders important questions in cell adhesion mechanics. One single AC is an intricate and dynamic compound of molecules. We hypothesize that an improvement in the molecular characterization within the clutch models could describe more accurately the mechanics of cell adhesion. In this contribution, we aim at understanding how the intracellular forces are transmitted along the main molecules of the adhesion complex, and how they actively respond to force by modifying their conformations and binding states. Specifically, we focus on the adhesion chain formed by the actomyosin network, the talin and integrin molecules, and the ECM. We studied how force transmission and behavior of these molecules change when the ECM stiffness and the type of integrin change. And, also, how these changes at the molecular scale change the whole adhesion complex dynamics. To do so, we follow the original clutch hypothesis and develop a detailed, multi-scale computational model of the AC. Both models' approaches are described in detail in the Models section.

The results are organized as follows: First, we model a molecular chain made of a talin protein attached on top to an actomyosin network and on the bottom to a substrate with varying stiffnesses. We use Green's functions [76] to compute the deformation of the substrate around each ligand position. We take a previous model of the talin mechanics that considers the conformational changes and non-linear deformation of each domain [24]. This allows us to analyze the talin behavior, both mechanical and conformational, under a physiological pulling force and substrate rigidities. Then, we introduce integrin molecules between the ECM and talins. Each integrin molecule links extracellularly to the ECM and intracellularly to the talin heads. Then, we bundle together these individual molecular chains to form an AC (Fig 1). First, we analyze the adhesion behavior, including their mechanical behavior, binding and unbinding kinetics, conformational changes, and exposure of actin-binding sites in the talin rod, when crowded with $\alpha_5\beta_1$ and $\alpha_V\beta_3$ integrins and we then perform a sensitivity analysis on the integrin's parameter to analyze how adhesion crowded with a different type of integrins would behave. We finish our results section by analyzing the case of substrates with different ligands spacing.

## Models

### Description of previous clutch models

The clutch model integrates the mechanical response of the substrate and the clutches, which represent the molecular chains, under the contractile pulling of an actomyosin network. Each chain can bind at a constant rate and unbind as force builds up on them. This binding and unbinding process is associated with the attachment and break of integrins with the ECM. The model solves five main variables involved in cell adhesion behavior [43, 46]: the probability of the clutches being bound, $P_b$, the substrate displacement, $x_{sub}$, the clutch displacement, $x_c$, the averaged forces in the clutches, $F_c$, and the actin network velocity, $v$.

The actomyosin network pulls on the adhesion molecules thanks to the contractile forces of the myosin motors. An actin filament is bound to several myosin motors, $n_m$, that generate contractile forces in the molecular chain. The actin flow is simply computed by a force-velocity

relation as:

$$v = v_u \left( 1 - \frac{F_{sub}}{F_{stall}} \right),$$

(1)

where $v_u$ is the unloaded velocity of the actin flow. The retrograde flow velocity reduces if the ratio between the reaction force in the substrate $F_{sub}$ and the stall force of the myosin motors, $F_{stall}$, increases. In the case of mature ACs, thick stress fibers crowded by myosin motors pull on the adhesion patch with a total stall force $F_{stall} = n_m F_m$, where $F_m = 2pN$ is the force required to stall the activity of one myosin motor and $n_m = 40$ the number of motors in the system (see S1 Text for details).

Then, the displacement of the engaged clutches is computed as $\Delta x = v\Delta t$, where $\Delta t$ is the time step in the Monte Carlo or Gillespie simulation as described below. The force at each $i - th$ clutch, $F_{c,i}$, is given by

$$F_{c,i} = \kappa_c (x_{c,i} - x_{sub}),$$

(2)

where $\kappa_c$ is the stiffness of each molecular clutch and $x_{c,i}$ is the displacement of the $i - th$ molecular clutch. The substrate is represented by a simple Hookean spring such that the force on the substrate is $F_{sub} = \kappa_{sub} x_{sub}$. The stiffness of the linear spring is $\kappa_{sub} = E4\pi a/9$, where $E$ is the Young modulus of the substrate and $a$ is the radius of the AC. Then, the displacement of the substrate is obtained by solving the balance of forces between the $n_{eng}$ engaged molecular clutches and the substrate as:

$$x_{sub} = \frac{\kappa_c \sum_{i=1}^{n_{eng}} x_{c,i}}{\kappa_{sub} + n_{eng}\kappa_c}.$$

(3)

Then, we can obtain $F_{sub}$ and the cell traction as $P = F_{sub}/\pi a^2$, where ($\pi a^2$) is the area occupied by a circular AC.

Once the force at each clutch, $F_{c,i}$ is obtained, we compute the binding and unbinding events at each molecular clutch. The $n_c$ molecular clutches are allowed to associate with the ECM, with the rate $k_{on}$, or to disengage, according to a dissociation rate $k_{off}^*$. Bell's model is the simplest approach for a force-dependent unbinding rate that increases exponentially as

$$k_{off}^* = k_{off} \, e^{\left( F_{c,i}/F_b \right)},$$

(4)

where $k_{off}$ is the unloaded dissociation rate and $F_b$ is the characteristic bond rupture force. This law follows a slip behavior, meaning that as the force increases, the lifetime of the bond decreases exponentially (see Fig 2). Experimental data also showed bonds that follow a catch behavior [40], in which the dissociation rate first decreases and then increases exponentially with the applied force as

$$k_{off}^* = k_{off,slip} \, e^{\left( F_{c,i}/F_{b,slip} \right)} + k_{off,catch} \, e^{\left( -F_{c,i}/F_{b,catch} \right)}.$$

(5)

$k_{off,slip}$ and $k_{off,catch}$ are the unloaded dissociation rates and $F_{b,slip}$ and $F_{b,catch}$ are the characteristic bond rupture forces, via the slip and catch pathways, respectively. The new bound and unbound clutches are then updated for the next time step when all the relations above are again computed. The catch bond model in Eq 5 does not take into consideration the Cyclic Mechanical Reinforcement (CMR) [73]. The CMR increases the lifetime of integrins when

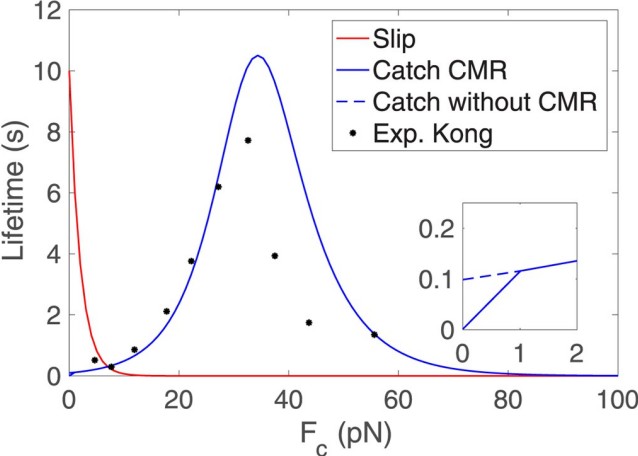

**Fig 2.** Lifetime of the weakest link for slip (red) and catch bonds with CMR (solid blue). Experimental data of integrin $\alpha_5\beta_1$ are shown in dots (black) [73]. The inset shows a zoom of the lifetime for small forces: catch case with CMR (solid blue) and without CMR (dashed blue).

they are subjected to cycles of engagement and disengagement before a stable AC is formed. To take into account the effect of CMR, previous models increased the off rate at low forces ($< 1$ pN) (see Fig 2) [8] and adopted an unbinding rate with CMR

$k_{off}^* = 8.10^4 \; e^{(F_c/8.16)} + 10.14 \; e^{(-F_c/6.24)} + 900 \; e^{(-F_c/0.01)}$. We will use this particular form of $k_{off}^*$ in this section. The results of the clutch model are discussed in the introduction and presented in section *Results of the clutch model* in the S1 Text.

The solution of the clutch model relies on repeated random sampling, which is usually solved by Monte Carlo (MC) or Gillespie methods. We use here MC simulations with constant time step, $\Delta t = 0.005$ s. During the simulation, many events of engagement and disengagement and, often, cycles of formation and rupture of ACs, occur. To evaluate the choice of the final time, we run MC simulations with different final times and choose $t_f = 100$ s as large enough to achieve accurate results when averaging over time the MC simulations and at the same time reduce the computational cost of the simulations (see Fig A6 in S1 Text for details).

## A multi-scale clutch model for adhesion complexes

Then, we take the modeling framework of the clutch model described above and extend it to incorporate a full-length model of the talin rod mechanics (see [24] for details) and the local deformation of the ECM due to each binding site.

The 13 domains of the talin rod have their specific unfolding and refolding rates and a non-linear force-displacement relationship (Fig 3). We assume that vinculin binds to the unfolded R3 domain, which induces integrins recruitment in the AC, as we described in the previous section. We also assume that actin binds to talin in the ending R13-DD part of the rod. The unfolding rates of the talin rod domains follow Bell's model and the folding rates follow an Arrhenius law [24, 77]. Mechanically, the folded domains behave as a freely-jointed chain (FJC), where the end-to-end distance of the folded domain, $x_{fol}$, is

$$x_{fol}(F) = l_0 \coth\left(\frac{Fl_0}{k_BT}\right) - \frac{k_BT}{F}. \tag{6}$$

$l_0$ is the rigid body size of the folded domain and $k_BT$ is the thermal energy. The unfolded

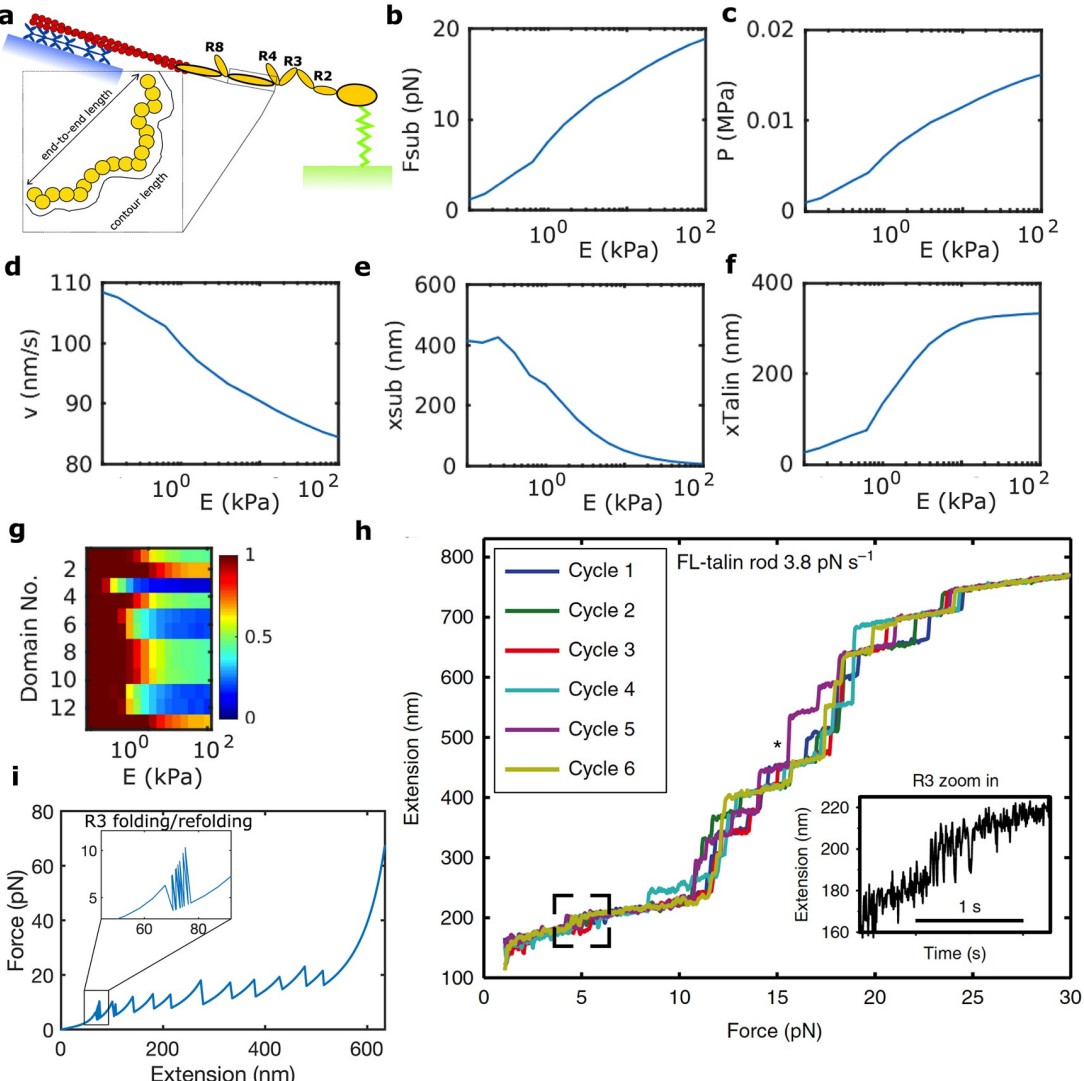

**Fig 3. Composition of a single molecular chain and role of the ECM rigidity on an adhesion chain.** (a) From bottom to top, a single adhesion chain made of a substrate (green), represented as a spring, with one end attached to a fixed surface and the other bound to talin (yellow). The talin's rod is made of 13 domains and its tail is attached to an actin fiber (red), which is connected to a number of myosin motors (blue), anchored to a fixed surface. Each talin's domain behaves as a WLC or an FJC in their unfolded and folded states, respectively. (b-f) Model variables $F_{sub}$, $P$, $v$, $x_{sub}$ and $x_{talin}$ for a molecular chain kept bound while varying Young's modulus E of the substrate in the range 0.1–100 kPa. (g) The color map shows the average folded (red) or unfolded (blue) state for each Young's modulus and domain of the talin rod. (h) An extension-force relation of the talin rod for a force-driven test at a rate of 3.8 pN / s. These results are reproduced from a previous publication [24]. (i) An force-extension relation of the talin rod in our model. The Young's modulus of the substrate is 100 kPa, so it can be compared to data by Yao et al. [24]. Each peak corresponds with the unfold of a talin domain (11 in total).

domains behave as a Worm-Like Chain (WLC),

$$\frac{FA}{k_B T} = \frac{1}{4(1 - x_{unf}/L_0)^2} - \frac{1}{4} + \frac{x_{unf}}{L_0}, \tag{7}$$

where $x_{unf}$ is the end-to-end length of the unfolded domain, $A$ is the persistence length and $L_0$ is the contour length. All talin model parameters are described elsewhere [24].

We use the binding and unbinding rates defined in Table 1 to model the integrin behavior for catch bonds. We use a linear spring to model the force-displacement relation in each integrin molecule, such that $F = k_c(x_c - x_{sub})$. $x_{int} = x_c - x_{sub}$ is the displacement of the integrin molecule and $x_c$ is the displacement at the integrin-talin bond position.

To mechanically model the extracellular space, we use Green's functions in a semi-infinite and isotropic medium [82]. The Green's function solution relates the elastic displacement in the i-direction at a point **x** due to a force in the j-direction applied at the origin as

$$G_{ij} = \frac{1}{16\pi\mu(1 - v)x} \left[ (3 - 4v)\delta_{ij} + \frac{x_i x_j}{x^2} \right], \tag{8}$$

where $\mu$ is the shear modulus, $v$ the Poisson's ratio of the medium and $x$ is the distance from the field point and the point of the force application.

We consider a 2D substrate on the XY-plane, where engagement points have coordinates **x** $= (x, y)$. The angle $\theta = 15°$ between the adhesion chains and the substrate [5, 12] gives forces in the same direction and results in a substrate displacement with non-zero components in the three dimensions. Here, we consider horizontal forces and substrate displacements laying in the XY plane. Moreover, we consider that all bound binders bear forces in the same direction as all actin filaments in the stress fiber and molecular chains bundle in parallel. Therefore our substrate displacements will have only one dimension and all chains pull on the ECM in the same direction. Therefore, the displacement of the substrate in the direction $i$ due to the force also in the direction $i$ is given as $x_{sub,i} = \hat{G}F_{c,i}$, where $\hat{G} = G_{ii}$ is the Green's function referring to the only relevant direction $i$.

**Table 1. Model parameters for an AC with $\alpha_5\beta_1$ and $\alpha_V\beta_3$ integrins.** For Young's modulus of the substrate, we take again the range $E = 0.1$–100 kPa. For the radius of the FA, we take a fixed value of $a = 708$ nm [50] with equispaced ligands at a distance of $d = 100$ nm. The number of equispaced ligands $n_c$ is then computed given the radius of the circular AC and the distance $d$, and we get $n_c = 158$ binders (see Fig A5 in S1 Text). For the stiffness of the linear spring that models the integrin, we take $\kappa_c = 10$ pN/nm [40, 73]. For the initial density of integrins, we choose $d_{int}^0 = 300$ int/$\mu$m$^2$ [8, 69]. The stall force of a single myosin motor is $F_m = 2$ pN [78] and we take several myosin motors equal to the number of ligands, following previous models [43, 69]. As for the unloaded actin velocity $v_u$, we choose $v_u = 110$ nm/s, similar to velocities measured in previous experimental data [79–81].

| Parameters | $\alpha_5\beta_1$ | $\alpha_V\beta_3$ |
|---|---|---|
| $k_{ont}$ ($\mu$m$^2$/s) | 0.005 | $1 \times 1^{-4}$ |
| $k_{off,slip}$ (s$^{-1}$) | $3.68 \times 10^{-4}$ | $4.17 \times 10^{-4}$ |
| $F_{b,slip}$ (pN) | 7.168 | 5.4825 |
| $k_{off,catch}$ (s$^{-1}$) | 2 | 0.4012 |
| $F_{b,catch}$ (pN) | 7.168 | 28.67 |
| $E$ (kPa) | 0.1–100 | |
| $a$ (nm) | 708 | |
| $d$ (nm) | 100 | |
| $n_c$ | 158 | |
| $\kappa_c$ (pN/nm) | 10 | |
| $d_{int}^0$ (int /$\mu$m$^2$) | 300 | |
| $F_m$ (pN) | 2 | |
| $n_m$ | 158 | |
| $v_u$ (nm/s) | 110 | |
| $k_{onv}$ (s$^{-1}$) | $1 \times 10^8$ | |
| $int_{add}$ (int /$\mu$m$^2$) | 24 | |
| $m_r$ (int /$\mu$m$^2$) | 15000 | |

We call $\boldsymbol{x}_{sub}$ the vector of one-dimensional displacements $x_{sub(s)}$ in all the points $s$ inside the FA, and we call $\boldsymbol{F_c}$ the vector of forces $F_{c(b)}$ in the ligand points $b = 1, \ldots, n_c$. Therefore, the displacement of the substrate at all sampling locations is given as $\boldsymbol{x}_{sub} = \hat{\boldsymbol{G}} \cdot \boldsymbol{F_c}$.

Once all model variables are solved, the traction exerted by the AC on the substrate is simply computed as $P = F/A$. As Green's function is singular at the point where the binder is bound and the force is applied, the displacement is computed at a distance of 5 nm from the binder. The simulations are run with the model parameters in Table 1.

To solve computationally the multi-scale clutch model, we use a Gillespie algorithm [83, 84] with variable time step, which is determined by the events rates: binding/unbinding of integrin molecules to the ECM and folding/unfolding rates of the talin rod. This allows us to track the integrin and talin dynamics in greater detail. The time at which each event, $i$, happens is computed as $t_i = -\ln \xi_i / k_i$, for all binders $i = 1 \ldots n_c$. $\xi_i$ are independent random numbers uniformly distributed over [0, 1] and $k_i$ is the event rate. After computing the time of all possible events at the current time step, we choose the minimum time $t_i$ and update the corresponding event. The link to the code is available in the S1 Text.

## Results

### Role of the ECM rigidity on a single adhesion chain

To understand the effect of the talin mechanics in the adhesion behavior, we extend previous analysis [24] to model a single adhesion chain where talin attaches to a substrate in a range of rigidities, $E = 0.1$–100 kPa from one side and remains bound at any time step. We don't include the integrin molecule between the ECM and talin. On the other end, talin links to an actin filament, that is also bound to $n_m$ myosin motors which are anchored to a fixed surface and generate a contractile force on the actin filament (see Fig 3a).

We run the model using as final time $t_f = 8$ s, which allows for most of the talin domains to unfold in any substrate rigidity. The force and tractions on the chain increase from zero to approximately 20 pN and 15kPa in the stiffest substrates, respectively (Fig 3b and 3c). The resisting force that myosin motors experience also reduces the velocity of the actin filament (Fig 3d). As the rigidity of the ECM increases, the total displacement imposed in the chain is absorbed by the substrate, $x_{sub}$, and by the talin, $x_{talin}$ (Fig 3e and 3f). Therefore, the elongation of the talin rod varies from approximately its resting length in the softest substrates (E = 0.1 kPa), where the substrate deforms at its maximum (400nm), to $\approx$350 nm in the stiffest ECM (E = 100 kPa), where the substrate effectively doesn't deform. Talin not only elongates but also unfolds and refolds over time (Fig 3g). At every unfolding event, the length of the talin rod further elongates. We found that, on average, most domains are folded for soft substrates (<10kPa). The unfolding events are mostly triggered starting from a substrate rigidity of $\approx$ 20 kPa. As we move toward stiff substrates (>30 kPa), the force increases and the domains unfold progressively, being the R3 domain the most likely one to unfold, followed by the R5, R6, R10, and R11. Importantly, we show forces and displacements of chain components, as well as the unfolding of each domain, which is in agreement with previous data (Fig 3h and 3i). Note, however, that our simulations are driven by displacements, so at each unfolding event we obtain a jump in force, while previous work was force-driven and they obtained jumps in displacement when a talin domain unfolded.

As we do not allow the chain to disengage, the displacement of the substrate $x_{sub}$ decreases while increasing Young's modulus of the substrate (Fig 3c).

## Role of the ECM rigidity in $\alpha_5\beta_1$ and $\alpha_V\beta_3$-crowded focal adhesions

Next, we focus on ACs crowded with either $\alpha_5\beta_1$ or $\alpha_V\beta_3$ integrins, among the most ubiquitous types of integrins. The lifetime curves together with the experimental data for $\alpha_5\beta_1$ [40] and $\alpha_V\beta_3$ integrins [8] are shown in Fig 4. Because $\alpha_5\beta_1$ integrins undergo CMR, we increase its lifetime to reflect the fact that integrins that have been subjected to loading and unloading cycles have a longer lifetime [73]. We do not consider CMR for $\alpha_V\beta_3$ because, as far as we know, it has not been demonstrated for this type of integrin. We summarize all values for the model parameters in Table 1.

We show the response of the ACs crowded with $\alpha_5\beta_1$ and $\alpha_V\beta_3$ integrins as a function of the substrate rigidity in Fig 5. The cell traction increases with the ECM stiffness, reaching a maximum of $\approx 115$ Pa and $\approx 100$ Pa for ACs crowded with $\alpha_V\beta_3$ and $\alpha_5\beta_1$ integrins, respectively, at $E = 10$ kPa and remain constant above 10 kPa. The two types of integrins analyzed, with two different lifetime landscapes, deliver similar cell tractions. These results indicate that although integrins' behavior may change the dynamics of the AC and its mechanosensing response, the resulting cell tractions may not be altered. The velocity decreases with the increase of the substrate rigidity because of the opposing force to the actin network movement. The minimum velocity, which is again approximately constant above 10 kPa, is $\approx 60$ nm/s and $\approx 50$ nm/s for the $\alpha_V\beta_3$ and the $\alpha5\beta1$ adhesions, respectively. These traction and velocity values are in agreement with previous clutch models and experimental results [8, 69] (see Fig 5 and discussion in the introduction Section).

Then, we explore the nanoscale of the model, which provides information on the molecular behavior (Fig 6). The percentage of bound binders increases as the stiffness of the substrate increases, reaching maximum values of 49% and 16% for the $\alpha_5\beta_1$ and $\alpha_V\beta_3$ cases, respectively, at $E = 100$ kPa. The lower percentage for the $\alpha_V\beta_3$ case is due to the lower binding rate of this type of integrin. The maximum force reached the bound binders, $F_{c,max}$, increases from 10–12 pN at 0.1 kPa to 20 pN at 1 kPa in both types of integrins and remains constant until 100 kPa. This is in agreement with force values reported at single talin and integrin molecules [24, 40, 73]. The displacement induced by the myosin motors in the molecular chains is mostly absorbed by soft substrates, while integrin and talin molecules absorb the displacement imposed when the substrate increases above the stiffness of these molecules. The maximum

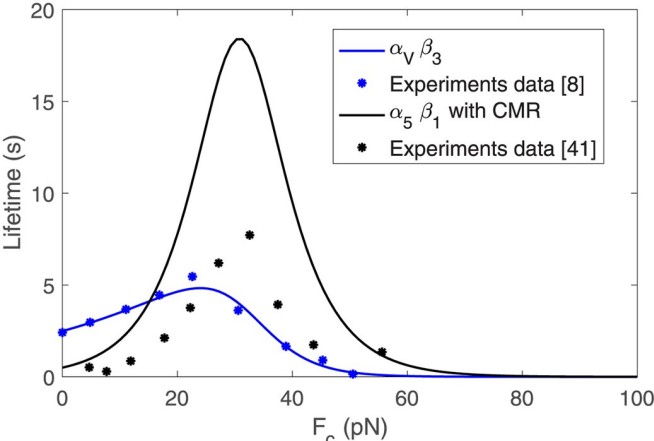

**Fig 4. Lifetime for $\alpha_5\beta_1$ integrins with CMR and for $\alpha_V\beta_3$ integrins.** The parameters obtained for $k_{off}^*$ are reported in Table 1.

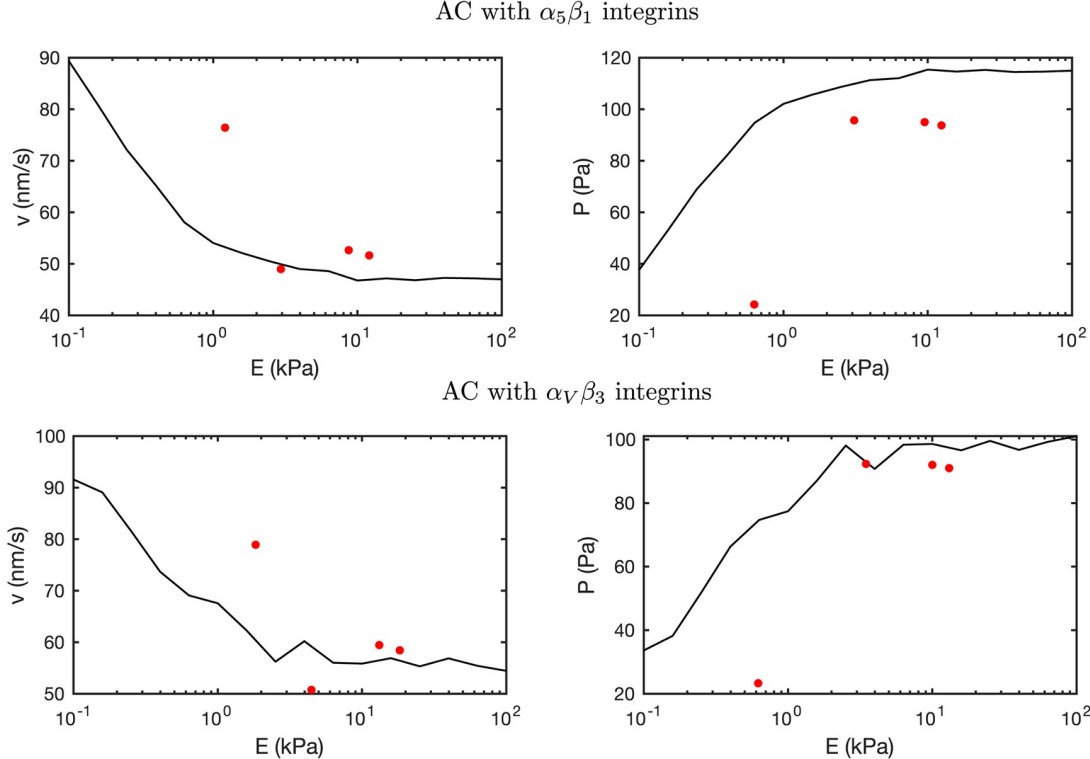

**Fig 5. Experimental and computational comparison of AC with $\alpha_5\beta_1$ and $\alpha_V\beta_3$ integrins.** The computational results of tractions and retrograde flow velocities are averaged over 10 Gillespie simulations and plotted against Young's modulus of the substrate $E$. The model results are compared against previous experimental data [8], represented with red dots.

displacement of both integrins is $\approx 2nm$, which is in the order of magnitude of previous experimental results [40, 73]. Similarly to other model variables, the displacement of talin increases until substrates of $\approx 10$ kPa, and then it remains constant up to the stiffest substrates at $\approx 400$ nm and $\approx 440nm$ for the $\alpha5\beta1$ and the $\alpha_V\beta_3$ cases, respectively. These results are also in close agreement with previous experimental data of the force-displacement relation in the talin rod [24]. Partially, this difference in the talin rod elongation is due to a higher number of unfolded domains in the $\alpha_V\beta_3$ case. Consequently, the probability of finding a bound binder with a vinculin molecule attached is higher in the $\alpha_V\beta_3$ case because there are more unfolded R3 domains, to which vinculin attaches to. Strikingly, the overall number of vinculin molecules in the AC is larger in the $\alpha5\beta1$ case (18) because the number of bound binders is larger.

In summary, our results for the tractions and actin flow velocity, which represent the macroscopic variables of the model, are similar to the reinforced case shown in Fig A2 in S1 Text, and to previous experimental data [8]. These are important results because previous clutch models closely reproduce and predict experimental data of these quantities. Moreover, our results also show a close agreement with the results of forces and displacements at individual talin and integrin molecules during cell adhesion [24, 40, 73], which represents a clear step forward in the modeling and understanding of the mechanosensing of ACs.

## Adhesion dynamics in $\alpha_5\beta_1$ and $\alpha_V\beta_3$-based adhesion complexes

To better understand the behavior of ACs crowded with $\alpha_5\beta_1$ or $\alpha_V\beta_3$ integrins, we analyze the evolution of a single Gillespie simulation in time. We focus on a substrate rigidity of E = 2.5

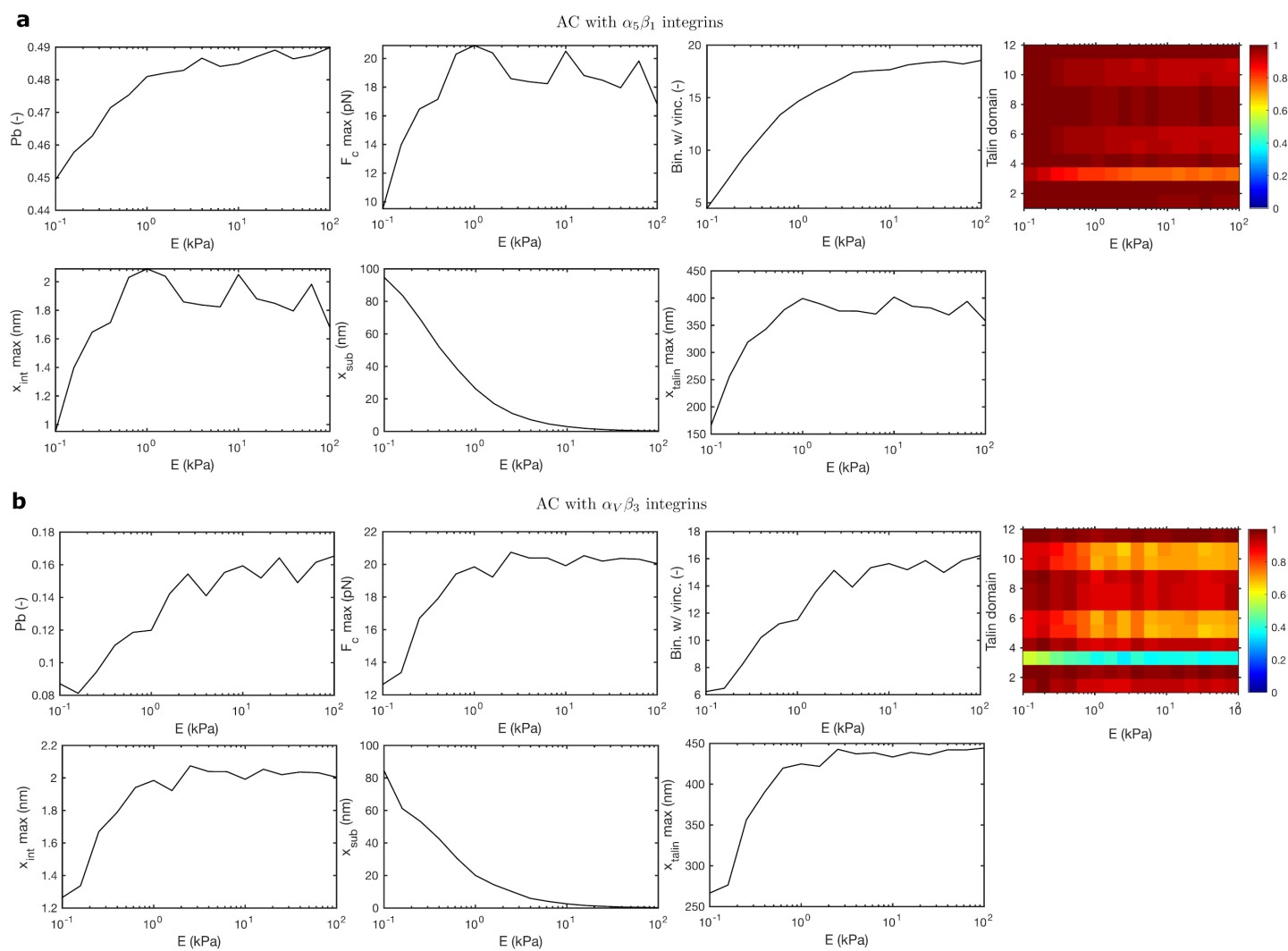

**Fig 6. Computational results in an AC with $\alpha_5\beta_1$ and $\alpha_V\beta_3$ integrins.** Results are averaged over 10 Gillespie simulations. We plot $P_b$, Bin. w/ Vinc., folding/unfolding states, $x_{int}max$ and $x_{talin}max$ and $F_cmax$ against Young's modulus of the substrate $E$ for ACs crowded with $\alpha_5\beta_1$ (a) and $\alpha_V\beta_3$ (b). The color plot shows the folded/unfolded state of talin domains and it is obtained for one Gillespie simulation, averaging over the bound binders.

kPa (see section *Supplementary figures* in the S1 Text for results on different substrate rigidities). The model parameters are the same as in the previous section.

For the $\alpha_5\beta_1$ case (Fig 7a), the adhesion enters a quasi-static phase in which the percentage of bound binders stabilizes to $\approx 50\%$ after a short period of $\approx 3$ s of transition from the initial free state (Fig 7a). The maximum force reached by a single binder is $\approx 15$ pN and presents a peak of 50 pN after which the integrin attachment breaks (Fig 7b). The maximum displacement of integrins is $\approx 6$ nm (Fig 7c). In terms of the talin dynamics, we show a very rapid landscape of folding and unfolding events that occur every few *ms* (see Fig 7d and 7e and section *Supplementary figures* in the S1 Text). As we showed before, R3 is the domain with more unfolded domains, $\approx 40\%$, followed by R5, R6, R10 and R11, $\approx 20\%$. We also see a maximum talin displacement of $\approx 600nm$, which represents a value close to talin fully unfolded contour length, although this is again only achieved once during the simulation time. All these results are again in agreement with previous data on integrins [40, 73] and talin [24].

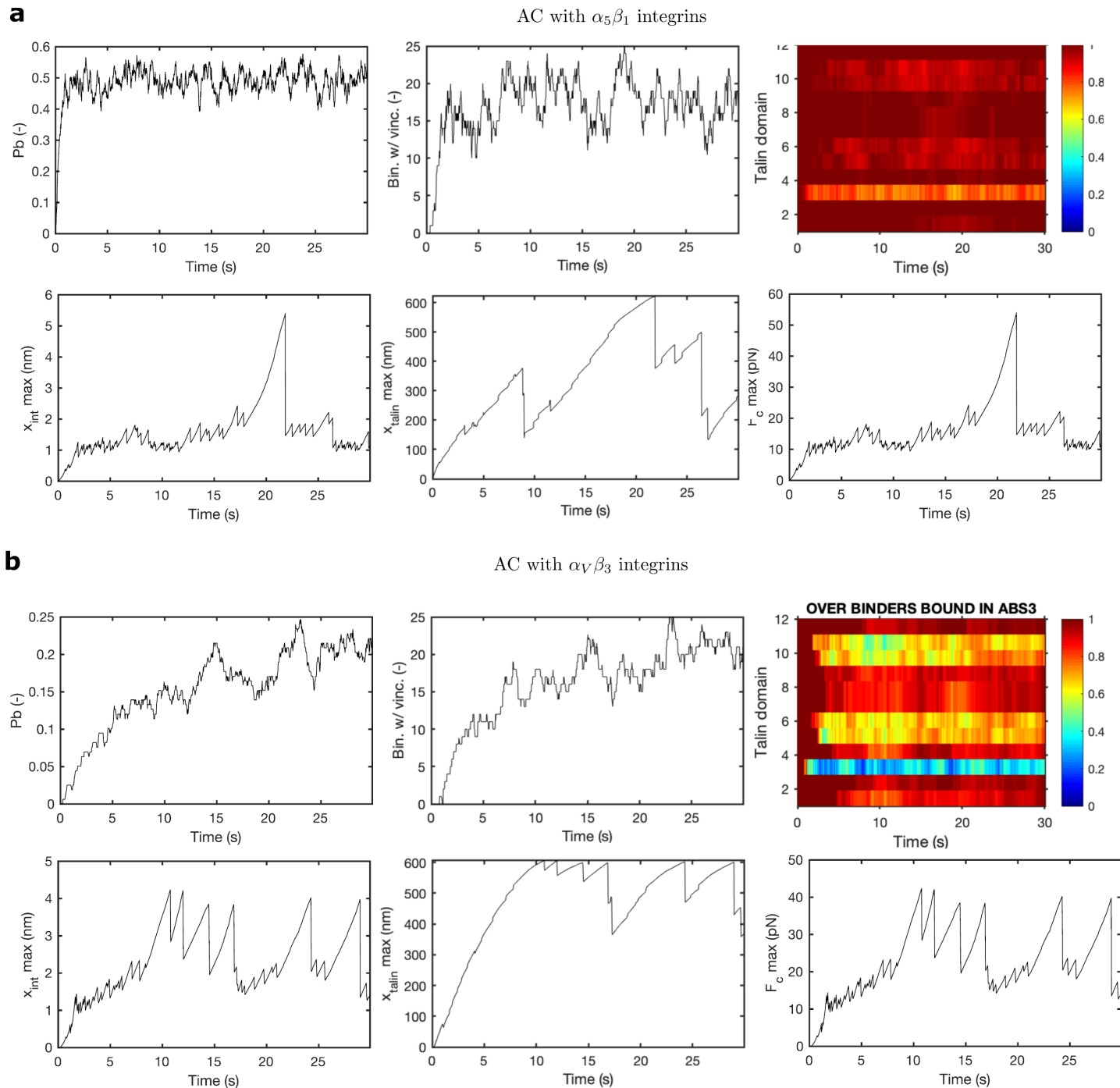

**Fig 7. Time behavior of AC with $\alpha_5\beta_1$ and $\alpha_V\beta_3$ integrins.** Results are for one Gillespie simulation, fixing Young's modulus of the substrate to $E = 2.51$ kPa. The figure shows the time evolution for $P_b$, Bin. w/ Vinc., folding/unfolding states, $x_{int}max$ and $x_{talin}max$ and $F_cmax$ for ACs crowded with $\alpha_5\beta_1$ (a) and $\alpha_V\beta_3$ (b).

The results for the $\alpha_V\beta_3$ case have similar behavior but present some differences (Fig 7b). First, the initial transition phase between the free state and the quasi-static state is $\approx 15$s, when clutches reach a $P_b \approx 0.16$ (Fig 7b). This is because the larger bond's lifetime cannot compensate for the lower binding rates. The force and integrin displacement peaks reach $\approx 40$ pN and

$\approx 4$ nm (Fig 7b), respectively, which are lower than in the $\alpha_5\beta_1$ case. These differences are due to the larger lifetime of $\alpha_V\beta_3$ integrins below $\approx 20$ pN, which also explains the larger number of peaks around that force value. The talin rod also shows a larger amount of unfolding events in all the talin domains than in the $\alpha_5\beta_1$ case (Fig 7). This is again because of a larger lifetime of the integrins below $\approx 20$ pN, which are force values at which most of the talin domains are unfolded. Indeed, we see again that the fully unfolded state at a displacement of $\approx 600$ nm, is reached several times during the simulation time. Again, the forces and displacements that reach each CAM agree with previous data [24, 40, 73] (Fig 7).

In short, the mechanosensitivity of both ACs crowded with these integrins gives counterintuitive results. The $\alpha5\beta1$ case has a larger amount of bound binders, which increases the available adhesion chains for vinculin binding, but there is a limited number of unfolded talin domains. On the other hand, the $\alpha_V\beta_3$ case has approximately half of the molecular chains available for vinculin binding. However, there is a larger amount of unfolded domains in the talin rod, that fosters a large number of vinculin attachments. As a result, both cases present a similar increase in vinculins bound.

## Integrins behavior in AC dynamics

Besides $\alpha_5\beta_1$ and $\alpha_V\beta_3$ integrins, there are several integrins implicated in cell adhesion, e.g. $\alpha_4\beta_1$, $\alpha_{IIb}\beta_3$, $\alpha_V\beta_6$ and $\alpha_V\beta_8$, with a diverse effect in the AC behavior and, therefore, in the mechanosensing of the cell [26]. Even for one specific integrin type, its behavior can change dramatically for different activation states, depending on the presence of different ions [40] or the level of CMR [73]. Consequently, its binding and unbinding rates can be altered and we can use these alterations to precisely engineer the adhesive of cells.

To explore the effect of different integrin types on the AC behavior, we performed a series of simulations by simultaneously varying the substrate rigidity and the integrin behavior. The range of parameters related to the integrins' behavior and their effects on the lifetime of the bond are shown in Fig 8. The other model parameters are kept as in previous sections (Table 1).

We can gather the effects of integrins model parameters in two groups. An increase of $F_b$, $k_{off,slip}$ or $k_{off,catch}$ results in a decrease of the bound binders and force transmission, which reduces cell traction (Fig 9b–7d). The decrease in the transmitted force is followed by a reduction of talin stretch and vinculin attachments. This behavior is more evident for variations of $k_{off,catch}$. These results are expected as we are enhancing the rupture of the bond at low forces by increasing these three parameters. We see the opposite response when we decrease the value of these three parameters. On the other hand, when we increase $k_{on}$, we make the bonds engagement faster than the disengagement and the behavior is analogous to a decrease in $F_b$, $k_{off,slip}$ or $k_{off,catch}$ (Fig 9a). The effect of increasing the ECM stiffness for all integrins model parameters is an increase in the talin displacement, in the number of vinculin molecules bound to the talin rod and tractions forces for any change of the integrins model parameters (Fig 9). The effect of the variations of the parameters in the traction is similar to the previous clutch models (see Fig 9 and Fig A8 in S1 Text).

Our results suggest that the on/off rates, which are indicative of different integrins and different activation states, result in very different ACs behavior in terms of molecular mechanosensing but also in the degree to which myosin motors can exert forces on the ECM.

## Variations in AC behavior due to ligand spacing

Finally, we investigate the effect of ligands spacing, or attaching locations, in the behavior of ACs [50]. We compute the ligand spacing as a function of the number of ligands in the

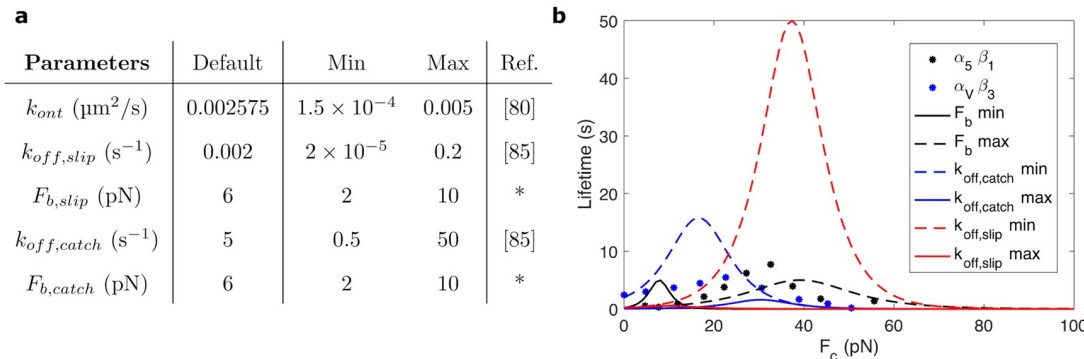

**Fig 8. Values of the model parameters and bond behavior.** (a) Default values and ranges for the parameters in the binding and unbinding rates of integrins for the multiscale clutch model. * indicates well-characterized parameters for which we only analyze a small range of values. For $k_{off,slip}$ and $k_{off,catch}$ we adopt an exponential distribution to cover several orders of magnitude. We use a linear distribution for the other parameters. We use previous experimental data on $\alpha_{IIb}\beta_3$ integrins to define the binding rate [85]. As for the range of the unbinding rate $k_{off}^*$, we take into account the experimental data for the lifetime of $\alpha_5\beta_1$, $\alpha_V\beta_3$ and $\alpha_L\beta_2$ integrins [86]. (b) Integrin lifetimes obtained with the extremes of the ranges for $F_{b,slip} = F_{b,catch}$, $k_{off,catch}$ and $k_{off,slip}$, together with experimental data for $\alpha_5\beta_1$ [40] and $\alpha_V\beta_3$ [8].

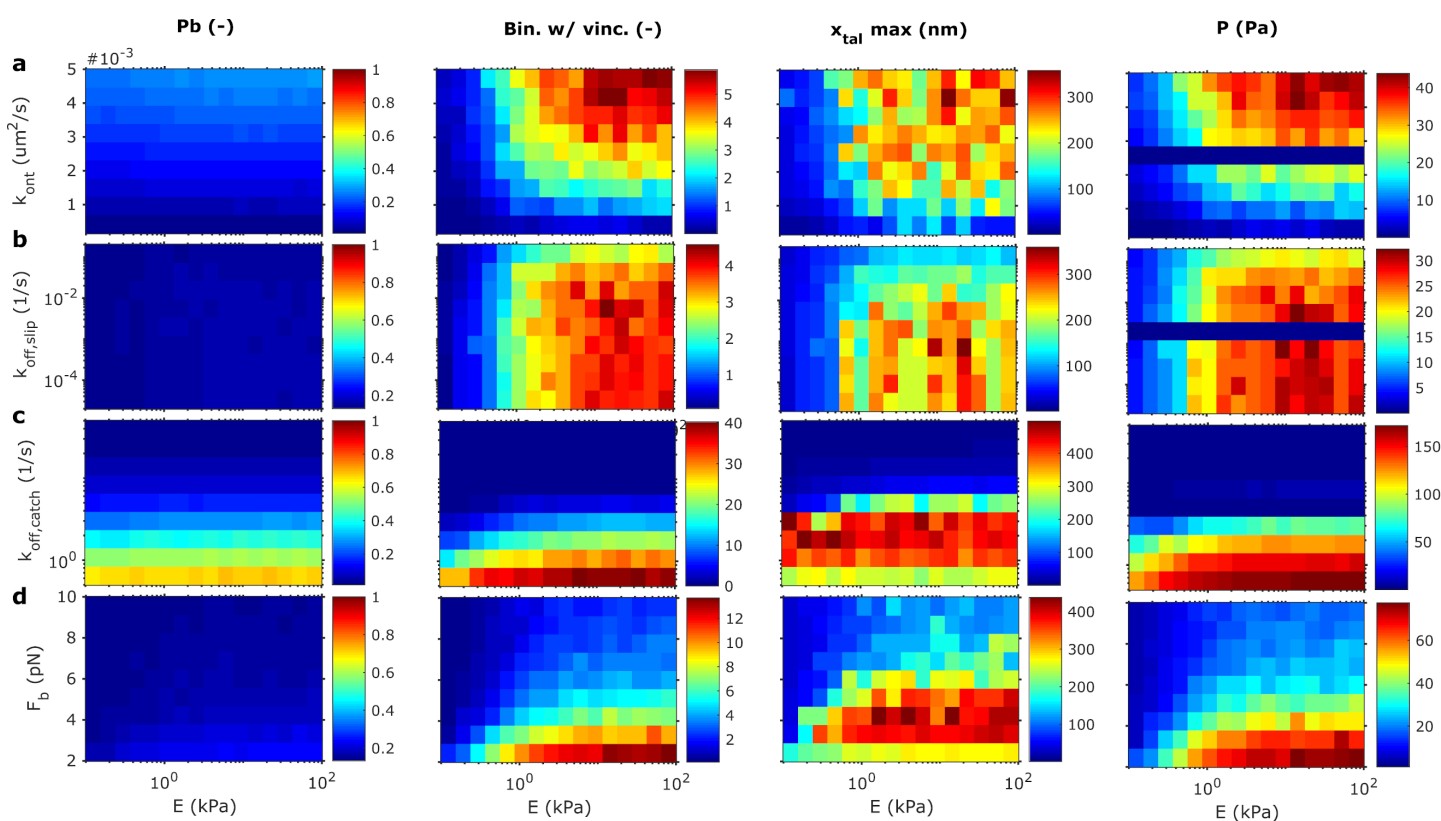

**Fig 9. Sensitivity analysis of the parameters integrin-fibronectin bond model.** Sensitivity analysis for the model parameters $k_{ont}$, $k_{off,slip}$, $k_{off,catch}$ and $F_{b,slip} = F_{b,catch}$ (a-d). We plot the variables $P_b$, the number of binders with vinculin (Bin. w/vinc), $x_{tal}^{max}$, and $P$ against Young's modulus of the substrate $E$ in columns. The results are obtained averaging over 10 Gillespie simulations.

substrate and of the AC radius (see Fig A5 in S1 Text for details). We focus on ACs crowded with $\alpha_5\beta_1$ integrins and use experimental data of the adhesion length as a function of the substrate stiffness for ligands placed 50, 100, and 200 nm apart [50]. These data and the resulting number of ligands are summarized in Fig 10. We analyze again the evolution of ACs for stiffness of the substrate in the $0.1 - 100$ kPa range. The remaining parameters are the same as in previous sections.

Our simulations show that the bound probability, $P_b$ (and $v$, $F_{c,bound}$, $x_{int,bound}$, $x_{talin,bound}$, and $d_{int,norm}$, see section *Supplementary figures* in the S1 Text) is similar for the three distances (Fig 11a). Therefore, the total number of ligands bound changes due to changes in ligand spacing for the same adhesion area or changes in the adhesion area for constant ligand spacing. Therefore, we obtain larger substrate displacements, $x_{sub}$ as we increase the number of ligands per AC (Fig 11b). $x_{tal}^{max}$ and $F_c^{max}$ increase as the stiffness of the substrate increases, with slightly higher values for substrates with ligand spacing of 50 nm than of 100 nm (Fig 11c and 11d). However, when the ligands are placed 200 nm apart, there is a sharp increase of these variables between 0.5 kPa and 2 kPa, overpassing the values for the 50 nm case at 2 kPa. Between 2 kPa and 10 kPa, where the adhesion size reduces to 180 nm and just 5 ligands, we obtain a sudden decrease in these variables. Above 10 kPa, these three variables keep constant. These results suggest that, on average, the behavior of the CAMs does not depend on the ligand spacing, although the maximum force and displacements of integrins and talins show striking differences depending on ligand spacing. Likewise, the total number of bound vinculins depends on the number of bound binders as well as on the number of unfolded domains per talin (Fig 11e). Consequently, we show an increase of bound vinculins up to 20 and 80 bound vinculins in the 50 nm and 100 nm spacing, respectively, at maximum substrate stiffnesses. However, the 200 nm case does not present bound vinculins in stiff substrates because of the very low number of bound binders and because, of these few bound binders, none of them have sufficient talin domains unfolded for vinculin to bind.

Cell traction increases for increasing ligands distances and substrate rigidities (Fig 11e). The maximum tractions, found for the stiffest substrates, are $\approx 75$, 150, and 600 Pa for the 200 nm, 100 nm, and 50 nm cases, respectively, at the maximum ECM stiffness. The averaged force per bound binder is similar for the three distances, therefore traction in the ACs only depends on the total number of ligands bound. This is achieved either by increasing ligands density or by increasing AC size for a fixed distance. These results indicate that the actual

**a**

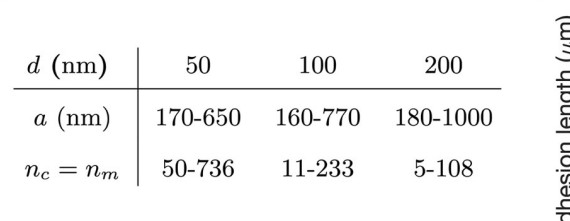

| $d$ (nm) | 50 | 100 | 200 |
|---|---|---|---|
| $a$ (nm) | 170-650 | 160-770 | 180-1000 |
| $n_c = n_m$ | 50-736 | 11-233 | 5-108 |

**b**

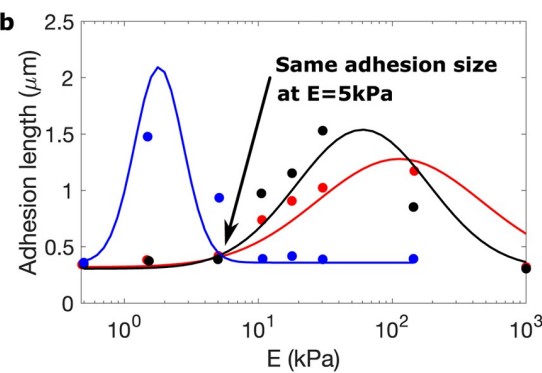

**Fig 10. Cell adhesion size as a function of the distance between ligands.** (a) Size of the ACs [50] and fit of the data to a Gaussian hill for ligand spacing of 50 nm (red), 100 nm (black), and 200 nm (blue). (b) The number of ligands $n_c$ and radius of the adhesion $a$ for the variation of ligand spacing in $\alpha_5\beta_1$ integrins.

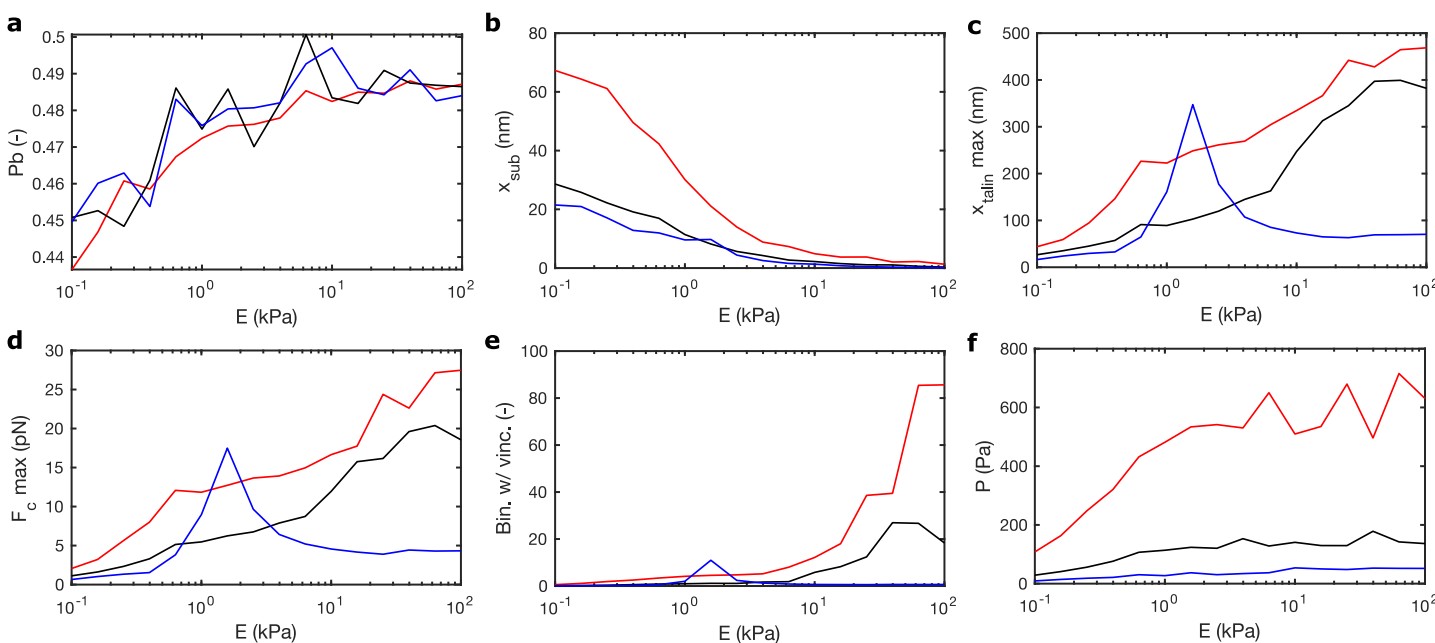

**Fig 11. Results for variation of ligands spacing for $\alpha_5\beta_1$ integrins.** (a-f) Results show $P_b$, $x_{sub}$, $x_{tal}^{max}$, $F_c^{max}$, Bin.w/ vinc. and $P$ for distance between ligands of $d$ 50 nm (in red), $d = 100$ nm (in black) and $d = 200$ nm (in blue).

number of bound binders is the determining factor for cell traction and that the behavior of the single adhesion chains, and their mechanosensing differences, may function to establish constant cell traction for ACs of the same size and equal ligand spacing.

## Discussion

Integrin-based cell adhesions are responsible for the attachment of cells to the ECM [1, 3] and, therefore, are central to the cell function. Among others, they control cell motility in development, e.g. during neural formation [87], nuclear mechanotransduction [88], and diseases processes such as tumor invasion [89].

In this work, we have first carefully analyzed previous clutch models for cell-ECM dynamics. As previously shown [8, 43, 69], they closely reproduce the cell response at the whole cell scale. However, we identified several inconsistencies in the mechanical response at the scale of the CAMs. However, the clutch models were not derived to describe this scale of the system. Those inconsistencies arise from a simplified description of the AC. First, the chain of CAMs was reduced to a unique linear spring. This is not just a structural simplification but, more importantly, it results in unphysiological values of the forces and displacements at the molecular level. Second, the substrate is represented by a single linear spring element that gathers the deformation of the entire AC. A group of adhesion chains would link to several attachment points in the ECM and, therefore, the deformation exerted on the ECM should change pointwise.

To address these issues, we modified previous clutch models and introduced the actual mechanical and conformational behavior of the molecules that form the mechanosensitive chain in cell adhesions. We used a talin model [24] that includes the non-linear mechanical behavior, the unfolding and refolding events of each talin domain, and the subsequent vinculin binding to the talin rod. We also introduced the integrin's behavior [40, 73]. Finally, we modeled the ECM deformation at each binding location using Green's functions. We then

integrated all these elements in previous clutch models and solved the adhesion dynamics at the whole AC as well as the individual CAM scale.

First, we simulate one single adhesion chain made of a talin molecule attached to a contractile actomyosin network from one side and to the ECM from the other. We used this model to validate the simplest building block of our model. The results were in agreement with previous results [22, 24]. Then, we put together many adhesion chains, or clutches, together to form an adhesion complex. We used our multi-scale model to understand how ACs crowded with either $\alpha_V\beta_3$ or $\alpha5\beta1$ behave mechanically at both the AC and the molecular levels. Our results agree with previous experimental data on traction forces at the cell scale (see Fig 5). More importantly, our results on the deformation and force in each CAM closely also agree with previous data (see the discussion on the force distribution in integrin-based cell adhesions in the introduction), which further validates our model. We summarize the comparison of experimental data and computational results from previous and current computational models of the force and displacement at each CAM in Fig 12. Still, whether the model exactly reproduces the behavior at the molecular scale in the context of a complete adhesion complex will have to be verified in the future. Then, we investigated how cell adhesion would change if other types of integrins would crowd the adhesion complex. This analysis allowed us to specifically pinpoint how traction force would change in different cell types. This is an important aspect of cell function because changes in cell traction are associated with alterations in cell function, including tissue cohesiveness or cell migration. Again, the results at the molecular scale will have to be verified in the future.

Experimental data on single ACs will be helpful to validate the model and advance toward a better understanding of AC behavior. Specifically, data on vinculin binding, force sensors in the talin and integrin molecules, integrin density, and AC size while varying the substrate stiffness would confirm the model predictions. However, simultaneous measurements of these quantities in each CAM within an AC are very complex experimental tasks and, as far as we know, have not been accomplished yet. We believe this is an important strength of our model because it can provide information in spatial and temporal details that no current experimental technology can deliver.

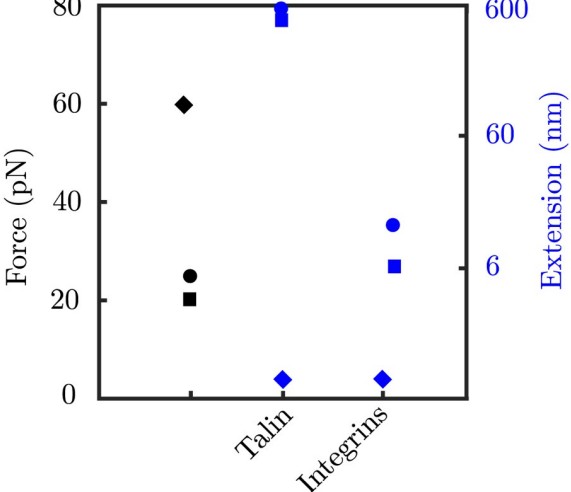

**Fig 12. Experimental and computational comparison of forces and extension of the molecular chains.** Experimental data is presented in circles (see the introduction for references), computational results from previous clutch models in diamonds, and current computational results in squares. Forces are represented by the symbols in black and displacement in blue. Displacements are separated for talin and integrins.

Finally, we analyze the effect of ligands' distance in cell behavior. We show that ligands' distance may not be responsible for changes in CAMs response. These differences are due to changes in the number of bound binders in the AC. However, previous data on cell tractions for $d$ = 50 nm and $d$ = 100 nm in human breast myoepithelial cells showed similar traction [50]. The discrepancy between the model results and the experimental data may be due to the spatial scales analyzed. Indeed, we have mostly compared our computational results with experimental data at the whole cell scale (see, e.g., [8, 43, 50]). However, our model specifically tackles single ACs while the experimental traction forces are usually obtained as an averaged quantity over cell sections larger than the actual AC size. The density of ACs per area, and not only the size of the AC, may explain these differences. For example, a higher density of AC for $d$ = 100 nm than for $d$ = 50nm would provide results comparable to previous experimental data [50].

We believe that we have advanced in the modeling and understanding of single ACs. However, there are still modeling aspects to improve for a better understanding of the cell adhesion behavior. For example, epithelial cells follow a catch bond behavior with talin reinforcement [8, 69]. Neurons, on the other hand, follow a pure slip behavior [43]. Although we have performed a computational sensitivity analysis, how specifically the type of integrins within the AC impacts the adhesion response across multiple cell types is still poorly understood. We had to increase integrins lifetime 2-fold with respect to the actual values [40] to reproduce the traction forces on ACs of cells crowded with $\alpha 5\beta 1$. This inconsistency could be the result of an incorrect model of the integrin dynamics or missing mechanotransductive aspects. Our model also predicts that force itself cannot be responsible for AC disassembling because, at a steady state, the rate of binding and unbinding reach a situation of semi-equilibrium in most cases. AC disassembling maybe then a downstream mechanotransductive result that we did not include in the model. Our model could be also used to study cell adhesion behavior changes for diverse affinities of different $\beta$-integrin tails for talin binding, the effect of other integrin binds sites in the talin molecule [36], or the mechanical behavior of vinculin along the adhesion chain [23]. Future studies should also focus on how forces induce mechanotransduction and biochemical processes to control AC disengagement. In line with this, we have considered that the AC remains always of the same size or, if we change it, it is directly imposed in the model based on previous experimental data. A more detailed analysis of the change in size during the entire formation and disengagement of the AC should also be addressed in the future.

Overall, we believe that our model is a step forward in the efforts of rationalizing cell adhesion mechanics. The model may also help us to engineer the cell adhesion response to design better biomimetic tissues [90–94] or to propose strategies to arrest tumor cell invasion [89].

## Supporting information

**S1 Text. Results of the clutch model.** Number of myosin motors and stall force. Extended data availability. Fig A1. Effect of slip and catch bonds in cell adhesion behavior. Fig A2. Effect of reinforcement by talin unfolding in cell adhesion. Fig A3. Timeevolutionoftheslipcase. Fig A4. Timeevolutionofthecatchcase. Fig A5. Number of equispaced ligands nc as a function of the adhesion radius. Fig A6. Comparison of the number of MC simulations for two different final times tf = 100s and tf = 1000s. Fig A7. Mean and standard deviation of the force Fc (pN) computed over bound binders. Fig A8. Sensitivity analysis of the reinforced case. Table A1. Model parameters for slip and catch cases. Table A2. Parameters for myosin motors used inside Eqs. 1, 2 and 3, with experimental references.
(PDF)

## Acknowledgments

We thank Pere Roca-Cusachs for the discussion.

## Author Contributions

**Conceptualization:** Chiara Venturini, Pablo Sáez.

**Formal analysis:** Chiara Venturini, Pablo Sáez.

**Investigation:** Chiara Venturini, Pablo Sáez.

**Methodology:** Chiara Venturini, Pablo Sáez.

**Supervision:** Pablo Sáez.

**Validation:** Pablo Sáez.

**Writing – original draft:** Chiara Venturini, Pablo Sáez.

**Writing – review & editing:** Chiara Venturini, Pablo Sáez.

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
