## [Decision Letter · Decision Letter 0]

1 May 2023

Dear Dr Saez,

Thank you very much for submitting your manuscript "A multi-scale clutch model for adhesion complex mechanics" for consideration at PLOS Computational Biology.

As with all papers reviewed by the journal, your manuscript was reviewed by members of the editorial board and by several independent reviewers. In light of the reviews (below this email), we would like to invite the resubmission of a revised version that takes into account the reviewers' comments.

One of the reviewers, in particular, suggested that the manuscript should more clearly explain the rationale behind its assumptions and modeling choices and the relevance of the model results for the open the questions the model tries to answer. The same reviewer also was concerned about a lack of references to experimental data and a lack of clarity in the figures, as well as some linguistic issues.

We cannot make any decision about publication until we have seen the revised manuscript and your response to the reviewers' comments. Your revised manuscript is also likely to be sent to reviewers for further evaluation.

Sincerely,

Martin Meier-Schellersheim

Academic Editor

PLOS Computational Biology

Jason Haugh

Section Editor

PLOS Computational Biology

Reviewer's Responses to Questions

**Comments to the Authors:**

Reviewer #1: In this paper, Venturini et al propose a new computational model in order to evaluate the contribution of single adhesion proteins to the clutch behavior. This is an important and unanswered question in the field of Integrin-based adhesions. The proposed computational model is based on on a mechanical system comprising elements mimicking integrin, talin, actin and a substrate of variable stiffness. These elements interact with one another governed by kinetic rates and transmit stresses and deformations which, in turn, strengthen or weaken the relative interactions.

Topic and question

-While the main idea behind this work is important and the exact interplay of several adhesion proteins to the clutch behavior is currently unknown, the study does not seem to pose and answer a clear scientific question and/or hypothesis regarding the complexity of mechanisms involved.

-The authors state: “We hypothesize that an improvement in the molecular characterization within the clutch models could answer these questions” . However, the five questions that the model wants to address are somehow vague, therefore it’s hard to understand the significance of the results. As an example, the question “How do individual molecular chains within the AC behave in different substrate stiffnesses?” does not refer to any specific protein (talin, integrin, or vinculin) and does not specify a specific type of behavior (kinetic rates, conformational change, exposure of binding sites, force transmission, unfolding). Therefore, once all proteins are put together into one model, it still remains unclear to me what is the exact question regarding their interplay. Second, how this model improvement is intended is not explained clearly. The method is presented at the end of the paper and some details are missing. For example, the authors state that they use Monte Carlo or Gillespie simulations. Which results are produced with one method and which results are produced with the other method? Also, why they use two methods? I invite the authors to focus the scope of the simulations and reorganize the results in order to answer a more specific scientific question, that can be directly discussed in the context of previous knowledge. I also invite the authors to explain their methods in more details and present it before the results.

Further questions about the explanation of the methods:

-at the end of pang. 17, the authors state:  dt is the time step in the Monte Carlo or Gillespie simulation(see Section ). What section?

-Regarding how the elements in the model are treated, are the engaged versus disengaged clutches considered separately in the simulations? is the force distributed only on the engaged clutches or on all clutches?

Results

-It’s hard to follow what is the main message of each figure. For example, in Fig. 8 several threshold values for combinations of substrate rigidity, force and and rate constants are identified. Are they meaningful? can these be related to any experimental observations or previous models?

-On page 16, the authors state: “our results on the deformation and force in each CAM closely also agree

with previous data”. Here a reference is needed. The reader doesn't know what data is this sentence referring to.

-Some predictions of the model have not been and cannot be experimentally tested. This limitation put into question the validity of the model itself. I invite the authors to more clearly list what results are validated and what results are predictions for future studies. For the predictions, I suggest making a case for why these results should be true based on indirect evidences in the field.

Writing and presentation

The manuscript needs language editing. The SI is missing several equation numbers through the text.

Reviewer #2: Review is uploaded as an attachment too.

The manuscript “A multi-scale clutch model for adhesion complex mechanics” by C. Venturini and P. Sáez develop a multi-scale computational for adhesion complexes based on the classical approaches of the clutch model. The model includes the typical parameters of the clutch models such as retrograde flow, myosin motors but two new ingredients (integrins and talin) are deeply analyzed. The multi-scale model dissects the role of integrins and talin at the molecular level, providing information about forces are transmitted through single chains. On top of the common proxies of the molecular clutch model as tractions and retrograde flow, the model shed light on the forces and elongation for integrins and talin. Authors start meticulously by studying the role of talin in the absence of integrins. After, they show how different integrins behave and how parameters that are unique to each type of integrins such as binding/unbindings rate can impact the force transmission in single chains and ultimately determine rigidity sensing. Finally, they use the model to study how the ligand spacing impacts force transmission and adhesion formation. The manuscript is well-written, the data is carefully presented, and the model results are validated and in good agreement with previous experimental results. The parameters of the model are well explained, and the assumptions are supported by the current literature.

While this reviewer understands the simplicity of the model, I would like to raise two points that it would be interesting if the authors can discuss.

1. Talin presents four binding sites for integrins in its dimeric form.

2. Different beta integrin tails may show diverse affinities for talin binding (Mol. BioSyst., 2014, 10, 3217)

Can the authors speculate how the model results are affected by that? Can these variables be introduced in the model?

**Have the authors made all data and (if applicable) computational code underlying the findings in their manuscript fully available?**

Reviewer #1: **No: **The code is not available

Reviewer #2: **No: **I don't see any statement in the manuscript where the make the code and data available.

PLOS authors have the option to publish the peer review history of their article (what does this mean?). If published, this will include your full peer review and any attached files.

Reviewer #1: No

Reviewer #2: No
---

## [Decision Letter · Decision Letter 1]

7 Jun 2023

Dear Dr Saez,

We are pleased to inform you that your manuscript 'A multi-scale clutch model for adhesion complex mechanics' has been provisionally accepted for publication in PLOS Computational Biology.

Please have a look at the suggested references provided by the reviewers and incorporate them where you see fit.

IMPORTANT: The editorial review process is now complete. PLOS will only permit corrections to add the references, correct spelling, formatting or significant scientific errors from this point onwards. Requests for major changes, or any which affect the scientific understanding of your work, will cause delays to the publication date of your manuscript.

Best regards,

Martin Meier-Schellersheim

Academic Editor

PLOS Computational Biology

Jason Haugh

Section Editor

PLOS Computational Biology

Reviewer's Responses to Questions

**Comments to the Authors:**

Reviewer #1: The paper has improved, but I still recommend trying to focus it more before publication.

Can the authors add a reference to the sentence: ‘integrins can reach a total length of 50nm”?

Reviewer #2: The manuscript “A multi-scale clutch model for adhesion complex mechanics” by C. Venturini and P. Sáez has been mostly revised according to the recommendations of the reviewers. Especially the comments from reviewer 1 about the goal of the model, the explanation of the model andf the clarity of the manuscript. While it was already clear to me, the manuscript has improved with the new structure and the editing. The comments of this reviewer (Reviewer 2) didn’t require any big changes in the manuscript and were merely brought up with the purpose to broaden the discussion with the authors and potentially to be discussed in the Discussion section.

Regarding the four potentially sites for integrin binding I do share with the authors some publications.

https://journals.biologists.com/jcs/article/130/15/2435/56321/Talin-the-master-of-integrin-adhesions.

‘Talin has two integrin-binding sites, three actin-binding sites, and can dimerise (see below and Fig. 2); therefore, each talin homodimer can potentially bind four integrin heterodimers and multiple actin filaments within this core structure.’

https://www.ncbi.nlm.nih.gov/pmc/articles/PMC3061283/

‘Notably, talin is a dimeric, integrin-binding protein that has 4 potential integrin binding sites.31 Thus, we suggest that there is a discrete multimolecular adhesion complex involved in spreading and adhesion that requires the juxtaposition of at least 4 RGD-liganded integrins within 60 nm, and involves talin.’

This reviewer understands and accepts author’s response about the affinity of talin for different integrins tails. I’d be great if this is implemented in future improvements of the model.

**Have the authors made all data and (if applicable) computational code underlying the findings in their manuscript fully available?**

Reviewer #1: Yes

Reviewer #2: Yes

PLOS authors have the option to publish the peer review history of their article (what does this mean?). If published, this will include your full peer review and any attached files.

Reviewer #1: No

Reviewer #2: No

---

## [Editor Report · Acceptance letter]

10 Jul 2023

PCOMPBIOL-D-23-00264R1 

A multi-scale clutch model for adhesion complex mechanics

Dear Dr Saez,

I am pleased to inform you that your manuscript has been formally accepted for publication in PLOS Computational Biology. Your manuscript is now with our production department and you will be notified of the publication date in due course.

With kind regards,

Zsofi Zombor
